# Stage-Specific Effects of TiO_2_, ZnO, and CuO Nanoparticles on Green Microalga *Haematococcus lacustris*: Biomass and Astaxanthin Biosynthesis

**DOI:** 10.3390/md23050204

**Published:** 2025-05-11

**Authors:** Ludmila Rudi, Tatiana Chiriac, Liliana Cepoi, Svetlana Djur, Ana Valuta

**Affiliations:** Institute of Microbiology and Biotechnology, Technical University of Moldova, MD-2028 Chisinau, Moldova; tatiana.chiriac@imb.utm.md (T.C.); liliana.cepoi@imb.utm.md (L.C.); svetlana.djur@imb.utm.md (S.D.); ana.valuta@imb.utm.md (A.V.)

**Keywords:** *Haematococcus lacustris*, metal oxide nanoparticles, life cycle, biomass, astaxanthin, lipids

## Abstract

Evaluating the effects of nanoparticles on biomass growth and astaxanthin accumulation in *Haematococcus lacustris* is crucial for optimizing the production of astaxanthin, a valuable carotenoid with numerous industrial applications. Identifying the life stages at which these nanoparticles exert stimulatory or toxic effects will aid in formulating effective production strategies. This study investigated the effects of titanium dioxide (TiO_2_), zinc oxide (ZnO), and copper oxide (CuO) nanoparticles on biomass growth, astaxanthin biosynthesis, and lipid accumulation in *Haematococcus lacustris*, with a focus on their stage-specific impact throughout the algal life cycle. The nanoparticles were added at the start of cultivation, and the microalgal cultures developed continuously in their presence. Sampling for biochemical analyses was performed at distinct life stages (green motile, palmella, and aplanospore), enabling the assessment of stage-dependent responses. TiO_2_NPs significantly stimulated biomass accumulation during the green motile stage. In the palmella stage, astaxanthin levels decreased in the presence of all nanoparticles, likely due to the absence of a stress signal required to activate pigment biosynthesis, despite ongoing biomass growth. In contrast, the aplanospore stage exhibited reactivation of astaxanthin biosynthesis and increased lipid accumulation, particularly under TiO_2_NPs. Astaxanthin content increased by 21.57%. This study highlights that TiO_2_, ZnO, and CuO nanoparticles modulate growth and astaxanthin biosynthesis in *Haematococcus lacustris* in a life cycle-dependent manner.

## 1. Introduction

Microalgae are photosynthetic organisms widely used in biotechnological applications because they produce biomass rich in bioactive compounds, such as carotenoid pigments, carbohydrates, lipids, and proteins [1,2]. Among them, the green microalga *Haematococcus pluvialis* stands out for its capacity to synthesize astaxanthin—a carotenoid with exceptional antioxidant properties—valued in the pharmaceutical, food, and aquaculture industries [3,4,5].

Various strategies have been developed to enhance astaxanthin production based on the microalga’s response to stress factors. Among the most effective methods are exposure to high-intensity light or specific wavelengths, with LEDs allowing precise control of the light spectrum [6,7]. Modifying the culture medium composition, especially by limiting nitrogen and phosphorus content, promotes astaxanthin accumulation through adaptive metabolic mechanisms [8,9]. Furthermore, supplementation with salts, metals, or pro-oxidant compounds can intensify oxidative stress, activating biosynthetic pathways in carotenoid production [10,11].

Recently, nanoparticles have emerged as a promising strategy for optimizing the production of valuable biomolecules. Metallic nanoparticles can act as stress inducers and as sources of essential trace elements, thereby influencing algal metabolism and the physiological response of photosynthetic cells [12]. Titanium dioxide (TiO_2_NPs), zinc oxide (ZnONPs), and copper oxide (CuONPs) nanoparticles have been investigated for their effects on various microalgal species, showing both beneficial and toxic effects depending on their morphology and concentration [13]. TiO_2_NPs can enhance photosynthetic efficiency by reflecting and absorbing light, thus promoting algal growth [14]. ZnONPs may provide zinc, an essential trace element for enzymatic activity in photosynthesis and carbon fixation processes. CuONPs, although less studied, have been associated with modulating oxidative and photosynthetic responses in a dose-dependent manner [15].

The interaction between nanoparticles and microalgae is complex and depends on factors such as the algal species, nanoparticle concentration, and specific cultivation conditions. In the case of the green microalga *Haematococcus pluvialis*, studies have reported both toxic and stimulatory effects induced by nanoparticles, depending on the type and concentration of nanomaterials and the growth conditions [16,17]. Some studies report growth inhibition and the accumulation of reactive oxygen species (ROS), leading to oxidative stress and disruption of algal metabolism [18]. On the other hand, certain nanoparticles used at appropriate concentrations have been shown to stimulate photosynthesis and pigment accumulation through mechanisms related to oxidative stress [19]. However, their influence on specific stages of the *Haematococcus pluvialis* life cycle, particularly on the transition from vegetative cells to red aplanospores rich in astaxanthin, remains poorly documented, requiring further investigation.

This study aimed to evaluate the effects of TiO_2_, ZnO, and CuO nanoparticles on biomass and astaxanthin accumulation in *Haematococcus lacustris*, with a particular focus on stage-specific responses during distinct physiological stages of its life cycle.

## 2. Results

### 2.1. Stage-Specific Biomass Response of Haematococcus lacustris to Metal Oxide Nanoparticles

Figure 1 shows the changes in *Haematococcus lacustris* biomass during the green motile stage of the life cycle under the influence of titanium, zinc, and copper oxide nanoparticles.

For all analyses, biomass was sampled at three representative time points: day 9 (green motile cell stages), day 13 (palmella stages), and day 16 (aplanospore stages), reflecting the major physiological stages of the microalga.

Exposure of *Haematococcus lacustris* to titanium oxide nanoparticles in the cultivation medium stimulated biomass accumulation during the green motile stage. TiO_2_NP concentrations that induced a statistically significant increase ranged from 1 mg/L to 20 mg/L, leading to biomass increases of 38.37 (*p* < 0.05)–39.32% (*p* < 0.01) compared to the control. At 0.1 mg/L, the increase was 15.10% (*p* < 0.05), while at 30 mg/L, the stimulatory effect was weaker, with only a 12.24% increase. ZnONPs enhanced biomass accumulation at concentrations of 20 mg/L and 30 mg/L, resulting in increases of 15.10% (*p* < 0.05) and 17.69% (*p* < 0.05), respectively. Concentrations between 1 mg/L and 10 mg/L had no significant effect, except for 0.1 mg/L, which led to an 11.29% (*p* < 0.05) increase in biomass. CuONPs exhibited a different pattern compared to ZnONPs. Concentrations of 0.1–1 mg/L induced biomass increases of 17.55% (*p* < 0.05) and 14.83% (*p* < 0.05), respectively.

Changes in *Haematococcus lacustris* biomass during the palmella and aplanospore life cycle stages under TiO_2_, ZnO, and CuO nanoparticles are shown in Figure 2.

During the palmella stage, the biomass of *Haematococcus lacustris* increased significantly in the presence of TiO_2_ nanoparticles, with the highest stimulatory effect of 57.29% (*p* < 0.01) observed at a concentration of 1 mg/L. At concentrations of 10 mg/L and 20 mg/L, biomass increases remained high at 53.96% (*p* < 0.01) and 54.60% (*p* < 0.01), respectively, compared to the control. Concentrations of 0.1 mg/L and 30 mg/L resulted in palmella biomass increases of 31.97% (*p* < 0.01) and 27.75% (*p* < 0.01), respectively. All concentrations of ZnONPs stimulated biomass accumulation during the palmella stage. The highest increases, 35.29% (*p* < 0.01) and 33.25% (*p* < 0.01), were recorded at 20 mg/L and 30 mg/L, while a concentration of 0.1 mg/L led to a 25.06% (*p* < 0.05) increase. CuONPs also exhibited a stimulatory effect on *Haematococcus lacustris* in the palmella stage across all tested concentrations. Low concentrations (0.1 mg/L and 1 mg/L) induced biomass increases of 35.55% (*p* < 0.01) and 31.71% (*p* < 0.01), respectively. Moderate increases, ranging from 15.09% (*p* < 0.01) to 22.25% (*p* < 0.05), were observed at concentrations between 10 mg/L and 30 mg/L.

In the aplanospore stage, the stimulatory effect of nanoparticles persisted. TiO_2_NPs increased final biomass levels by 20.70% (*p* < 0.001) to 19.87% (*p* < 0.001), with the highest values observed at 20 mg/L and 30 mg/L. ZnONPs induced a uniform increase in biomass content, ranging from 12.53% (*p* < 0.001) to 14.10% (*p* < 0.001), in proportion to nanoparticle concentration. CuONPs produced a moderate stimulatory effect at concentrations between 0.1 mg/L and 10 mg/L, with biomass increases of 6.13% (*p* < 0.01) to 11.65% (*p* < 0.01) compared to the control. However, at 30 mg/L, a reduction of 27.66% (*p* < 0.001) in final biomass content was recorded.

### 2.2. Astaxanthin Accumulation in the Palmella and Aplanospore Stages of Haematococcus lacustris

Figure 3 illustrates the effect of TiO_2_, ZnO, and CuO nanoparticles on astaxanthin accumulation during the palmella and aplanospore stages.

In the palmella stage, all types of nanoparticles significantly reduced the astaxanthin content in *Haematococcus lacustris* biomass. TiO_2_NPs led to a progressive decrease in astaxanthin content with increasing nanoparticle concentration. At 0.1 mg/L, the astaxanthin content was 16.08% (*p* < 0.001) lower than the control, and at 30 mg/L, the reduction reached 32.75% (*p* < 0.001). In the case of ZnONPs, a reduction of 10.66% (*p* < 0.05) in astaxanthin content was observed only at the concentration of 0.1 mg/L. Astaxanthin levels in the biomass did not differ significantly from the control for the other concentrations. CuONPs showed a similar effect to TiO_2_NPs. The most pronounced reductions in astaxanthin content—34.49% (*p* < 0.001) and 34.88% (*p* < 0.001)—were observed at 10 mg/L and 20 mg/L, respectively. Unlike TiO_2_NPs, CuONPs also led to astaxanthin reductions at lower concentrations of 0.1 mg/L and 1 mg/L, with decreases of 24.42% (*p* < 0.001) and 28.68% (*p* < 0.001), respectively.

In the aplanospore stage, the astaxanthin content either remained at control levels or showed a slight, nonsignificant increase. For TiO_2_NPs, astaxanthin content did not vary significantly with concentration, ranging between 3.39% and 3.49% of the biomass, representing increases of 7.52% (*p* < 0.001) to 10.82% (*p* < 0.01) compared to the control. For ZnONPs, concentrations between 0.1 mg/L and 10 mg/L did not significantly affect astaxanthin content in microalgal biomass. However, at 20 mg/L and 30 mg/L, reductions of 11.54% (*p* < 0.05) and 40.80% (*p* < 0.001) were recorded, respectively. CuONPs produced a similar trend to ZnONPs. At concentrations between 0.1 and 10 mg/L, the astaxanthin content in the biomass was either similar to or higher than that of the control. However, at 30 mg/L, a reduction of 42.86% (*p* < 0.010) in pigment content was observed.

### 2.3. Lipid Accumulation in Haematococcus lacustris Biomass Under Nanoparticle Exposure

Figure 4 presents the changes in lipid content in *Haematococcus lacustris* biomass during the palmella and aplanospore stages of development under the influence of TiO_2_, ZnO, and CuO nanoparticles.

TiO_2_NPs caused a significant decrease in lipid content in microalgal biomass accumulated during the palmella stage, with values ranging from 14.77% to 19.63%, compared to 24.02% in the control. The most significant reductions—38.44% (*p* < 0.001) and 36.13% (*p* < 0.01)—were observed at 0.1 mg/L and 20 mg/L, respectively. In the case of ZnONPs, the most substantial reductions in lipid content were recorded at 0.1 mg/L and 1 mg/L, with decreases of 24.57% (*p* < 0.001) and 26.59% (*p* < 0.01), respectively. Concentrations of 20 mg/L and 30 mg/L also decreased, with lipid levels ranging between 14.45% (*p* < 0.05) and 13.00% (*p* < 0.001). CuONPs had an even more pronounced effect than TiO_2_NPs and ZnONPs, with lipid content reductions ranging between 37.28% (*p* < 0.01) and 41.04% (*p* < 0.01).

Lipid content increased in most cases in the aplanospore stage. TiO_2_NP application led to a concentration-dependent increase in lipid accumulation, reaching a maximum of 43.28% at 20 mg/L—corresponding to a 30.18% (*p* < 0.01) increase compared to the control. ZnONPs induced a moderate increase in lipid content, with values rising by 17.82% (*p* < 0.001) and 15.27% (*p* < 0.01) at 1 mg/L and 10 mg/L, respectively. In contrast, higher concentrations of 20 mg/L and 30 mg/L significantly reduced lipid content by 16.73% (*p* < 0.001) and 20.36% (*p* < 0.001). CuONPs in the culture medium increased lipid content from 16.36% (*p* < 0.01) to 18.90% (*p* < 0.01) compared to the control. The lowest nanoparticle concentration caused a 26.90% (*p* < 0.05) increase in lipid content, whereas the highest concentration (30 mg/L) resulted in a 31.27% (*p* < 0.01) reduction.

Overall, *Haematococcus lacustris* exhibited stage-specific responses to TiO_2_, ZnO, and CuO nanoparticles, reflected in variations in biomass, astaxanthin content, and lipid accumulation.

## 3. Discussion

The results obtained in this study highlight the differential response of the microalga *Haematococcus lacustris* to TiO_2_, ZnO, and CuO nanoparticles, depending on the developmental stage. The application of these nanoparticles at the selected concentrations demonstrated that they are not toxic to the microalgal culture.

The nanoparticles exerted a stimulatory effect on microalgal biomass production, evaluated at synchronized sampling points throughout the life cycle, ensuring comparability of developmental stages between treated and control variants. A key observation is the persistent nature of this stimulatory effect throughout the different developmental stages of *Haematococcus lacustris*. This is supported by the biomass accumulation data, which showed a positive trend regardless of nanoparticle concentration, with a few exceptions. This effect is likely attributable to the specific properties of the nanoparticles.

TiO_2_ nanoparticles may enhance light absorption and photosynthetic activity in autotrophic organisms, thereby increasing photosynthesis rates [20]. The effects of ZnO nanoparticles on microalgae are associated with two mechanisms: the direct action of the nanoparticles themselves and an ionic effect resulting from Zn^2+^ ion release. These ions act as micronutrients involved in the function of key enzymes, such as carbonic anhydrase (for CO_2_ fixation) and superoxide dismutase (for neutralizing superoxide radicals) [21]. CuO nanoparticles can absorb photons in the visible spectrum, generating mobile charge carriers that may interact with algal cellular processes [22].

In this study, the impact of TiO_2_, ZnO, and CuO nanoparticles on the vital and biosynthetic activity of *Haematococcus lacustris* was investigated with a focus on the type of inoculum used, namely aplanospore cells. The germination process of the aplanospores, which lasted three days, was carried out in a mineral medium supplemented with nanoparticles at different concentrations. The successful transition of aplanospores to green motile cells was considered the first indicator of the non-toxic nature of the tested nanoparticles, providing a preliminary assessment of their effects on cellular development.

During the green motile stage of the life cycle, *Haematococcus pluvialis* cells exhibit intense metabolic activity, characterized by active biomass accumulation and cell division [8]. Exposure of the microalga to nanoparticles generally led to enhanced biomass growth, suggesting that, at this stage, the cells may utilize the nanoparticles either as auxiliary sources of micronutrients or as stimulators of photosynthetic processes. This stimulation may occur directly, through the induction of moderate stress, or indirectly, by altering the metabolic conditions.

A significant increase in biomass content was observed in cultures treated with TiO_2_ nanoparticles, which act as metabolic modulators, influencing both carbon fixation and nutrient assimilation. Through their adsorption properties, nanoparticles can enhance photosynthetic efficiency and the availability of essential nutrients, which may explain the significant biomass increase observed in our experiments. Various studies have shown that the effect of TiO_2_ nanoparticles is both concentration- and species-dependent. For instance, a concentration of 100 mg/L TiO_2_ nanoparticles significantly stimulated biomass accumulation in *Scenedesmus quadricauda* (Chlorophyta) (*p* = 0.024), whereas in *Stigeoclonium tenue* (Chlorophyta), the highest biomass increase was observed at 200 mg/L TiO_2_ nanoparticles (*p* = 0.038). In the diatom *Planothidium lanceolatum* (Bacillariophyta), exposure to 50–200 mg/L TiO_2_ nanoparticles resulted in moderate biomass stimulation (*p* < 0.05) [23].

Another study investigating the influence of TiO_2_ nanoparticles on the growth and metabolic activity of *Scenedesmus quadricauda* found that the toxic effect was related to particle size, with larger nanoparticles (200 nm) exhibiting no significant impact on growth or photosynthetic activity [14]. A moderate toxic effect of TiO_2_ nanoparticles on *Haematococcus pluvialis* was observed when small-sized particles (14.02 ± 2.18 nm) were applied at a concentration of 100 mg/L. Although these particles did not significantly inhibit growth, they were associated with a slight delay in biomass accumulation, which was harvested during the stationary phase up to the formation of aplanospores [24].

A stimulatory effect was also observed for copper oxide and zinc oxide nanoparticles. However, at higher concentrations (20 and 30 mg/L), both types of nanoparticles reduced the growth of *Haematococcus. lacustris*, suggesting that the initial stimulatory effect may have been caused by the onset of moderate stress, which temporarily enhanced photosynthetic processes. Nevertheless, excessive stress levels likely exceeded the cells’ adaptive capacity, inhibiting growth.

For ZnO nanoparticles, a clear growth-promoting effect has been reported at concentrations up to 1 mg/L, significantly stimulating the biomass of *Chlorococcum* sp. Higher concentrations (e.g., 8.1 mg/L) induced marked toxicity, reducing biomass by up to 50% [21]. It is assumed that the stimulatory effect of ZnONPs is due to the supply of Zn^2+^, an essential micronutrient in photosynthesis, while the toxic effect is associated with strong oxidative stress.

Another study reported a significant biomass increase in *Picochlorum* sp. exposed to nanoparticles, where TiO_2_NPs (10 mg/L) induced a 2.76-fold increase in biomass, and ZnO NPs (10 mg/L) led to a 4.93-fold increase compared to the control. The growth stimulation was attributed to the gradual release of metal ions from the nanoparticles, which were incorporated into cellular metabolism, enhancing enzymatic and photosynthetic activity [22].

Regarding the effects of CuONPs, most studies report toxic outcomes, which may be partially mitigated by stress tolerance mechanisms and strain-specific resistance [15]. A stimulatory or neutral effect has been observed in the cyanobacterium *Arthrospira platensis* exposed to CuONPs [13]. Accordingly, in the vegetative stage of *Haematococcus lacustris* cultures, the nanoparticles demonstrated a clear stimulatory effect on biomass production. In contrast, most previously reported studies involving the application of nanoparticles to *Haematococcus pluvialis* culture media have found either a neutral or negative impact on biomass accumulation [16,17,24,25].

In the palmella stage, *Haematococcus pluvialis* cells cease active division and reprogram their metabolism toward reserve accumulation. At this intermediate stage, astaxanthin biosynthesis begins, the encystment process is initiated, and the cells take on a brownish hue—a mixture of green and orange [8]. During this stage, biomass accumulation continues in most experimental variants, albeit at a slower rate compared to the motile stages, reflecting the metabolic shift from proliferation to survival and stress preparedness. The amount of aplanospore biomass increases and is significantly higher compared to the control.

One possible explanation is the conversion of green motile cells into palmellas, which were present in large numbers in experimental variants and quantified spectrophotometrically. A similar effect was previously reported with NaCl treatment, which accelerated the transition of green cells into palmellas [10].

Although the transition to the palmella stage is a natural physiological process, nanoparticle application appeared to enhance certain metabolic changes. As cells enter the palmella stage, the influence of nanoparticles does not vanish but continues to affect metabolic balance. Thus, although the rate of biomass accumulation is lower compared to the motile stage, it remains active, while astaxanthin synthesis is diminished. This suggests the absence of a sufficient stress signal required to fully activate the conversion of carotenoid precursors into astaxanthin. The low astaxanthin content observed may reflect the persistence of a favorable, non-stress-inducing environment, supported by the nutritional input supplied by the nanoparticles, which maintains the cells in a metabolically active state but not oriented toward astaxanthin biosynthesis.

Certain nanoparticles, such as iron or zinc, can function as essential micronutrients for photosynthesis and cell division, increasing their availability in the medium [16,21]. This nutritional contribution may enhance biomass accumulation even in the absence of a major stress signal.

It has been reported that treatments with ZnO nanoparticles at low concentrations can stimulate growth and chlorophyll accumulation during the vegetative stages of *Haematococcus pluvialis* cultures, without triggering significant astaxanthin biosynthesis [8]. An important observation in this study was the consistently low lipid content. Lipid reduction may be associated with thylakoid degradation and the initiation of astaxanthin accumulation [26]. Monitoring the accumulation of phenolic compounds in *Haematococcus pluvialis* biomass has also demonstrated nanoparticle-induced stress [24].

Studies on the application of nanoparticles in *Haematococcus pluvialis* cultivation have highlighted their role as stimulators of astaxanthin synthesis, an effect attributed to intracellular oxidative stress [16,19,25,27]. A stimulatory effect on both biomass accumulation and astaxanthin synthesis was reported following treatment with silver nanoparticles (AgNPs) of 10 and 20 nm in size [28].

In the aplanospore stage, characterized by the maximum accumulation of protective metabolites, nanoparticles induced significant changes in astaxanthin and lipid content. The accumulation of these compounds is crucial for cell survival under extreme conditions, and the results obtained confirm that astaxanthin biosynthesis is reactivated at this stage, compensating for the earlier decline observed during the palmella stage.

Comparative analysis suggests that the initial cellular response to nanoparticles during the green motile stage influences subsequent metabolism, sustaining and amplifying effects throughout the palmella and aplanospore stages. Although astaxanthin content remained low during the palmella stage, its biosynthesis resumed in the aplanospore stage. This observation suggests that the nanoparticles did not elicit a sufficient stress response during the palmella stage, with biosynthetic activity significantly intensifying as the cells transitioned to the aplanospore stage. Correlation analysis showed that astaxanthin synthesis during the aplanospore stage was not negatively affected, and pigment accumulation occurred in parallel with lipid accumulation. Lipid biosynthesis, essential for astaxanthin’s structural integration and stabilization, is tightly linked to carotenoid production [26]. In all experimental variants, a strong positive correlation was observed between astaxanthin and lipid content, with r^2^ = 0.966 for TiO_2_NPs, indicating a very strong relationship, and r^2^ = 0.820 and r^2^ = 0.902 for ZnONPs and CuONPs, respectively, also indicating very strong positive correlations (Appendix A).

These results demonstrate that nanoparticles’ effects are not transient but influence the entire life cycle of the microalga. The initial interaction with nanoparticles triggers a metabolic reconfiguration that propagates through all developmental stages, leading to specific alterations in biomass, astaxanthin, and lipid accumulation. The continuity of this response suggests a dynamic, stage-dependent mechanism, highlighting the need to optimize nanoparticle application based on their long-term effects on microalgal metabolism.

Accordingly, the data obtained reveal the complex impact of nanoparticles on the metabolism of the green microalga *Haematococcus lacustris*, affecting each developmental stage distinctly. During the green motile stage, nanoparticles stimulate biomass growth; in the palmella stage, they reduce astaxanthin accumulation; and in the aplanospore stage, they may facilitate the biosynthesis of protective metabolites.

The final astaxanthin yield was determined by two key factors: biomass accumulation during the green stage and the intensification of astaxanthin biosynthesis during the aplanospore stage. In this context, the amount of accumulated astaxanthin varied depending on the nanoparticle type, reaching a maximum of 121.6 mg/L in the presence of TiO_2_NPs, representing a significant increase of 33.5% compared to the control. These findings are consistent with previous observations indicating that enhancing biomass content during the green stage can contribute to a higher final astaxanthin yield [29].

Altogether, these observations suggest that nanoparticle use can be strategically optimized to guide microalgal metabolism toward more efficient production of bioactive compounds, offering valuable perspectives for microalgae-based biotechnologies.

## 4. Materials and Methods

### 4.1. Microalgae Culture

This study used the green microalga *Haematococcus lacustris* (*pluvialis*), strain CNMN-AV-05, maintained in the National Collection of Nonpathogenic Microorganisms at the Institute of Microbiology and Biotechnology, Technical University of Moldova. The culture was grown in a mineral medium with the following composition: NaNO_3_—0.3 g/L, KH_2_PO_4_—0.02 g/L, K_2_HPO_4_—0.08 g/L, NaCl—0.02 g/L, CaCl_2_—0.05 g/L, MgSO_4_∙7H_2_O—0.01 g/L, ZnSO_4_∙7H_2_O—0.0001 g/L, MnSO_4_∙5H_2_O—0.0015 g/L, CuSO_4_∙5H_2_O—0.00008 g/L, H_3_BO_3_—0.0003 g/L, (NH_4_)_6_Mo_7_O_24_∙4H_2_O—0.0003 g/L, FeCl_3_∙6H_2_O—0.0175 g/L, and EDTA—0.0075 g/L. Cultivation was performed at 26–28 °C under continuous illumination at 75 µmol m^−2^ s^−1^, with periodic agitation during the first ten days. Aplanospore biomass was used as the inoculum. To induce astaxanthin biosynthesis, the light intensity was increased to 150 µmol m^−2^ s^−1^ for an additional four days. Metal oxide nanoparticles were added directly to the mineral medium.

The life cycle of *Haematococcus lacustris* (pluvialis) strain CNMN-AV-05, starting from red aplanospores, comprises the following stages: germination (days 0–3), green motile cell stage (days 3–9), palmella stage (days 9–13), and red aplanospore stage (days 13–16).

### 4.2. Nanoparticle Source and Characterization

The nanoparticles used in this study were purchased from Sigma-Aldrich (Merck KGaA, Darmstadt, Germany). Copper oxide nanopowder (CuONPs; product code 544868) had a particle size of 50 nm, as determined by transmission electron microscopy (TEM). The zinc oxide nanoparticle dispersion (ZnONPs; product code 721077) had a particle size of <100 nm (TEM), while the titanium dioxide nanoparticles (TiO_2_NPs; product code 718467) had a particle size of 21 nm (TEM) and a purity of ≥99.5%.

To disperse the CuO and TiO_2_ nanoparticles and prevent agglomeration, nanoparticle suspensions in deionized water were subjected to ultrasonication at a frequency of 22 kHz and an ultrasonic power density of 7 W/cm^2^. To avoid overheating, the suspensions were sonicated in 30-s pulses with 15-s intervals between cycles for a total treatment time of 4 min.

### 4.3. Experimental Design

An inoculum suspension was prepared using *Haematococcus lacustris* culture in the aplanospore stage, adjusted to a concentration of 0.2 g/L dry weight. Three types of metal oxide nanoparticles were used: TiO_2_, ZnO, and CuO. For each type of nanoparticle, 18 Erlenmeyer flasks (100 mL) were prepared, each containing 50 mL of algal suspension.

Nanoparticles were added to obtain final concentrations of 0 mg/L (control), 0.1 mg/L, 1 mg/L, 10 mg/L, 20 mg/L, and 30 mg/L. Each concentration was represented by three flasks corresponding to the three sampling time points (days 9, 13, and 16).

The cultures were maintained under controlled laboratory conditions.

Biomass sampling was performed on days 9, 13, and 16, corresponding to the green motile cell stage, palmella stage, and red aplanospore stage, respectively. At each sampling point, one flask from each experimental variant was processed to ensure the correspondence between the physiological stage and the applied treatment. The control sample was processed in an identical manner.

The experiment was carried out in three independent biological replicates to ensure data reproducibility.

### 4.4. Biomass Collection and Quantification

The biomass concentration of *Haematococcus lacustris* was determined spectrophotometrically at 680 nm (green motile cells), 480 nm (palmella stage), and 565 nm (red aplanospores). These wavelengths were selected based on full-spectrum absorbance scans, where the respective maxima were experimentally identified for each cell type. A specific calibration curve was constructed for each developmental stage by correlating optical density at the corresponding wavelength with known biomass concentrations based on dry weight measurements.

The biomass, consisting of green motile cells, palmella cells, and aplanospores, was then separated from the culture medium by centrifugation at 4000 rpm for 5 min and resuspended in distilled water to a final concentration of 10 mg/mL.

Green cell biomass was subjected to repeated freeze–thaw cycles, while palmella and aplanospore biomass underwent acid hydrolysis before further analysis.

### 4.5. Hydrolysis of Haematococcus lacustris Cysts

Following centrifugation, 1 mL of 0.1 M hydrochloric acid was added to each biomass sample. The resulting suspension was incubated in a water bath at 90 °C for 10 min. After incubation, the samples were rapidly cooled under running cold water, and the acid was removed by decantation.

The hydrolyzed biomass was then washed with distilled water to eliminate residual acid. For this purpose, 6–7 mL of distilled water was added to the biomass, followed by gentle agitation and decantation, during which the cysts remained in the sediment. This washing step was repeated three times. After the final wash, the biomass was collected by centrifugation at 4100 rpm for 7 min.

### 4.6. Astaxanthin Extraction and Content Determination

Astaxanthin was extracted from 10 mg of acid-hydrolyzed *Haematococcus lacustris* biomass using 5 mL of 96% ethanol at room temperature under continuous agitation for 120 min. The ethanolic extract was separated from the biomass by centrifugation at 4100 rpm for 5 min.

Astaxanthin concentration was determined spectrophotometrically at 478 nm. The content was calculated using a calibration curve established with standard solutions of pure astaxanthin (≥97%, Sigma-Aldrich Chemie GmbH, Taufkirchen, Germany) prepared in 96% ethanol. The calibration curve exhibited a high degree of linearity (R^2^ ≥ 0.99), and the identity of astaxanthin in the extract was confirmed by its characteristic absorption peak at the same wavelength.

### 4.7. Lipid Determination

*Haematococcus lacustris* biomass remaining after astaxanthin extraction was used for lipid quantification. A total of 10 mg of biomass was mixed with 5 mL of chloroform–ethanol mixture (9:1, *v*/*v*). The extraction was carried out at room temperature under continuous agitation for 120 min. After extraction, the solvent was separated from the biomass, and the alcohol fraction was removed by washing with distilled water. The chloroform phase was dried over anhydrous sodium sulfate, and the solvent was subsequently evaporated at 40 °C.

The dried lipid extract was subjected to acid hydrolysis using concentrated sulfuric acid. For this purpose, 2 mL of sulfuric acid was added to the dried extract, and the mixture was heated in a water bath at 90 °C for 10 min. After cooling, 0.1 mL of the hydrolysate was added to 3 mL of phosphovanillin reagent (prepared by dissolving 1.2 mg of vanillin in 1.0 mL of 68% phosphoric acid). The colorimetric reaction was allowed to develop for 30 min at room temperature. Absorbance was measured at 520 nm, and lipid content was calculated using a calibration curve constructed with standard oleic acid solutions.

### 4.8. Statistical Analysis

All experiments were performed in triplicate, and biochemical assays were conducted under identical conditions. Data were analyzed using Microsoft Excel and are presented as mean ± standard deviation. Statistical significance between experimental groups was assessed using Student’s *t*-test, with *p* < 0.05 considered statistically significant. Pearson correlation analysis was performed to evaluate the relationships among the values of the biochemical parameters. The Pearson correlation coefficients are available in the Appendix A.

## 5. Conclusions

This study demonstrated that TiO_2_, ZnO, and CuO nanoparticles significantly influence the development of the green microalga *Haematococcus lacustris*, with effects depending on the life cycle stage. The tested nanoparticles did not exhibit obvious toxic effects at the applied concentrations.

In the green motile stage, nanoparticles stimulated biomass accumulation, with the most pronounced effect observed for TiO_2_NPs. During the palmella stage, biomass accumulation continued, supported by the influence of nanoparticles; however, astaxanthin synthesis was reduced, most likely due to the absence of a sufficient stress signal to activate the conversion of carotenoid precursors. In the aplanospore stage, astaxanthin biosynthesis was triggered, and pigment content increased in response to treatment with low concentrations of nanoparticles. High concentrations of ZnONPs and CuONPs led to a significant reduction in astaxanthin content in *Haematococcus lacustris* biomass.

The correlation between nanoparticle type, biomass accumulation, and metabolite biosynthesis indicates an adaptive metabolic response in the microalga. Nanoparticles triggered a persistent metabolic reconfiguration throughout the developmental cycle, affecting the synthesis and accumulation of astaxanthin and lipids. The optimal use of nanoparticles may enhance *Haematococcus lacustris* productivity and improve the efficiency of biotechnological processes based on this species.

Evaluating the effects of TiO_2_, ZnO, and CuO nanoparticles on biomass growth and astaxanthin accumulation in *Haematococcus lacustris* is essential for understanding how these nanomaterials influence microalgal physiological processes. *Haematococcus lacustris* is a valuable source of astaxanthin, a potent antioxidant carotenoid with applications in the pharmaceutical and food industries. Given the growing demand for astaxanthin, identifying factors that stimulate its production and understanding the effects of nanoparticles is crucial for optimizing large-scale cultivation and production.

Metallic nanoparticles can significantly impact algal metabolism by stimulating biochemical processes or causing oxidative stress. Assessing their effects across different life cycle stages will provide better insight into the optimal timing of their application. Identifying the developmental stages at which nanoparticles exert stimulatory or toxic effects will support the formulation of strategies for the effective use of nanomaterials to maximize astaxanthin production and enhance the efficiency of microalgal biotechnologies.

## Figures and Tables

**Figure 1 marinedrugs-23-00204-f001:**
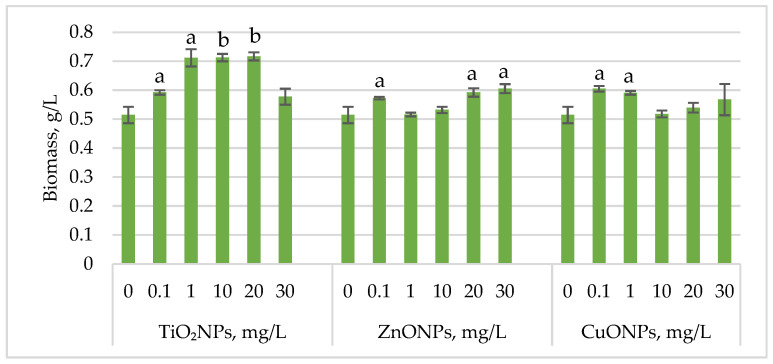
Biomass (g/L) of *Haematococcus lacustris* accumulated during the green motile stage under the influence of TiO_2_, ZnO, and CuO nanoparticles. 0—control variant; Statistical significance (Student’s *t*-test, each treatment compared to control, 0 mg/L): a—*p* < 0.05; b—*p* < 0.01. Values are mean ± SD (n = 3).

**Figure 2 marinedrugs-23-00204-f002:**
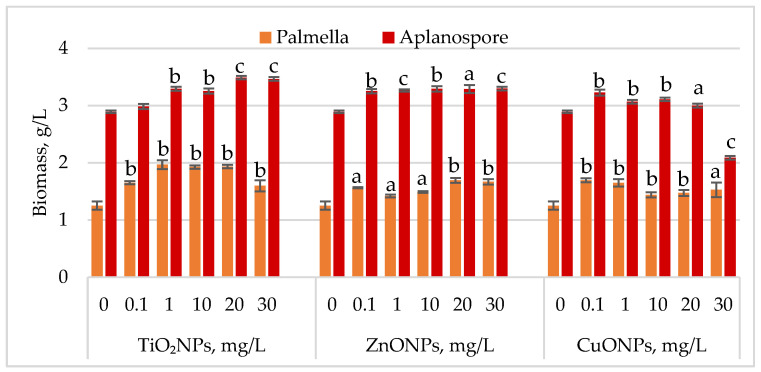
Biomass concentration (g/L) of *Haematococcus lacustris* accumulated during the palmella and aplanospore stages under the influence of TiO_2_, ZnO, and CuO nanoparticles. 0—control variant; Statistical significance (Student’s *t*-test, each treatment compared to control, 0 mg/L): a—*p* < 0.05; b—*p* < 0.01; c—*p* < 0.001. Values are mean ± SD (n = 3).

**Figure 3 marinedrugs-23-00204-f003:**
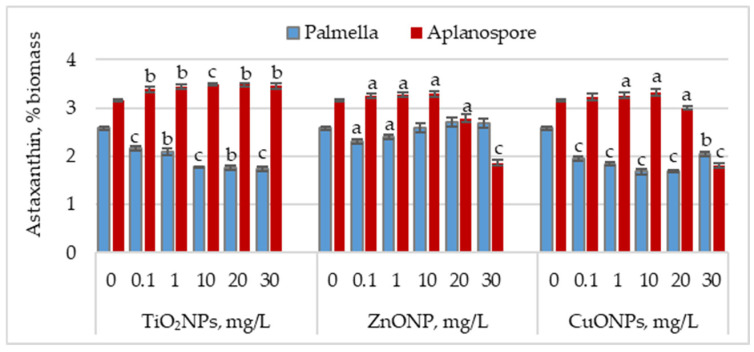
Astaxanthin content (% of dry biomass) in *Haematococcus lacustris* during the palmella and aplanospore stages under exposure to TiO_2_, ZnO, and CuO nanoparticles. 0—control variant; Statistical significance (Student’s t-test, each treatment compared to control, 0 mg/L): a—*p* < 0.05; b—*p* < 0.01; c—*p* < 0.001. Values are mean ± SD (n = 3).

**Figure 4 marinedrugs-23-00204-f004:**
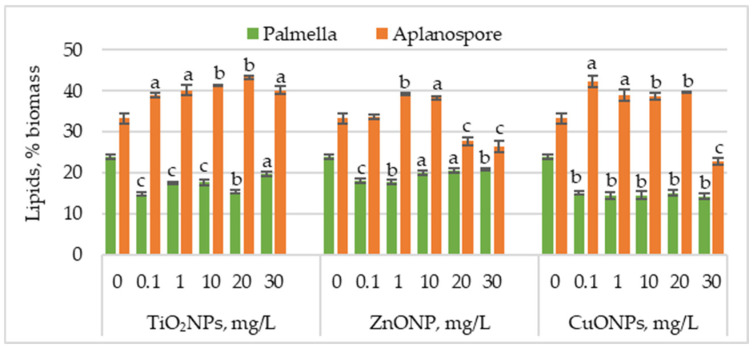
Lipid content (%) in *Haematococcus lacustris* biomass produced during the palmella and aplanospore stages under exposure to TiO_2_, ZnO, and CuO nanoparticles. 0—control variant; Statistical significance (Student’s t-test, each treatment compared to control, 0 mg/L): a—*p* < 0.05; b—*p* < 0.01; c—(*p* < 0.001). Values are mean ± SD (n = 3).

## Data Availability

The original contributions presented in this study are included in the article/Appendix A. Further inquiries can be directed to the corresponding author.

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
