# Peer review of "Stage-Specific Effects of TiO2, ZnO, and CuO Nanoparticles on Green Microalga Haematococcus lacustris: Biomass and Astaxanthin Biosynthesis"

_marinedrugs, 2025, doi:10.3390/md23050204_

Round 1
Reviewer 1 Report
Comments and Suggestions for Authors
REVIEW OF THE ARTICLE BY LUDMILA RUDI ET AL. ENTITLED ‘STAGE-SPECIFIC EFFECTS OF TiOâ‚‚, ZnO, AND CuO NANOPARTICLES ON GREEN MICROALGA HAEMATOCOCCUS PLUVIALIS: BIOMASS AND ASTAXANTHIN BIOSYNTHESIS’
The authors investigate the effect of varying concentrations of TiOâ‚‚, ZnO, and CuO nanoparticles on the growth and astaxanthin accumulation in the chlorophyte Haematococcus lacustris. Culture productivity was assessed by measuring biomass, and total lipid content was also determined. The classical biphasic cultivation strategy for H. lacustris was employed for astaxanthin production. The authors report some effects of the nanoparticles on microalgal parameters. Unfortunately, the current data presentation strategy and lack of appropriate statistical treatment hinder a critical evaluation of the conclusions and findings, particularly certain claims (e.g. line 183). Numerous similar studies have already investigated the effects of nanoparticles on Haematococcus and other microalgae. As such, the novelty of this work appears limited. Please find some specific critical comments below:
- H. pluvialis is an outdated and incorrect synonym; H. lacustris is the valid name (10.12705/652.11).
- The novelty is weak. There are several existing studies on the effects of TiOâ‚‚, ZnO, and CuO nanoparticles, including: 10.1016/j.jbiotec.2021.10.004, 10.7717/peerj.7582, 10.1142/S1088424624500664, and 10.1007/s10646-021-02406-5. Thus, the claim that this topic is poorly described (lines 63–66) lacks supporting evidence.
- The statistical treatment is inadequately described. Figure captions refer to ‘p<0.05; b – p<0.01; c – p<0.001’, but do not clarify the hypotheses tested or the specific comparisons made. In the Discussion (lines 310–313), correlation analysis is mentioned, yet no corresponding data or methodological description is provided.
- It is unclear from the Results section what exactly is represented by the bars in the plots. Do they depict differences between initial and maximum values? Or between initial and values obtained at a specific time point?
- The statements regarding astaxanthin ‘utilisation’ by algae (e.g. lines 21, 275, 280) should be reconsidered. Astaxanthin functions primarily as a passive photoprotective filter. There is currently no evidence for its degradation within algal cells. In some cases, a reduction in astaxanthin content as a percentage of biomass may reflect increased overall biomass, but this would require clearer explanation and detailed interpretation of the figures to support such a conclusion.
- The manuscript is poorly written, and the terminology used is often unclear. For instance, with respect to the haematocyst stage of the microalga, the terms ‘cyst’ and ‘aplanospore’ are used interchangeably, which adds to the confusion.
Author Response
Reviewer Comment: The authors investigate the effect of varying concentrations of TiOâ‚‚, ZnO, and CuO nanoparticles on the growth and astaxanthin accumulation in the chlorophyte Haematococcus lacustris. Culture productivity was assessed by measuring biomass, and total lipid content was also determined. The classical biphasic cultivation strategy for H. lacustris was employed for astaxanthin production. The authors report some effects of the nanoparticles on microalgal parameters. Unfortunately, the current data presentation strategy and lack of appropriate statistical treatment hinder a critical evaluation of the conclusions and findings, particularly certain claims (e.g. line 183). Numerous similar studies have already investigated the effects of nanoparticles on Haematococcus and other microalgae. As such, the novelty of this work appears limited.
Response: We thank the reviewer for this comment and the opportunity to clarify key aspects of our methodology and data interpretation.
All biochemical parameters (biomass, astaxanthin, and lipids) were measured in triplicate and expressed as mean ± standard deviation. Statistical significance was evaluated using Student’s t-test, comparing each nanoparticle concentration to the control (0 mg/L) within the same developmental stage (green, palmella, or aplanospore). The thresholds for significance (p < 0.05, p < 0.01, p < 0.001) were indicated in the figures, and the procedure is now described in more detail in the revised “Materials and Methods” section. To eliminate any ambiguity, we have also updated all figure captions to specify the statistical test applied.
Regarding the reviewer’s observation on line 183, the original sentence stating that “the nanoparticles exerted a stimulatory effect on microalgal biomass production” has been reformulated to clarify that this effect was evaluated at synchronized sampling points, ensuring comparability between treated and control cultures within the same developmental stage. As the study focused on stage-specific responses, the aim was not to compare values across life cycle stages but to assess nanoparticle dose–response effects within each physiologically distinct phase. This approach helped to avoid potential confounding from natural variation between stages in biomass, astaxanthin, and lipid levels.
Although statistical comparisons between stages could provide additional insights, such analysis was beyond the scope of this study, which was designed specifically to assess the stage-specific biological response to nanoparticle exposure.
Regarding the reviewer’s comment on the study's novelty, we acknowledge that several previous works have addressed nanoparticles' general effects—and especially toxicity—on microalgae. However, our contribution lies in the targeted analysis of Haematococcus responses to metal oxide nanoparticles at three physiologically distinct stages, using a synchronized cultivation strategy with and without nanoparticle exposure. The cultivation was initiated using red cysts (aplanospores) in mineral medium supplemented with nanoparticles, allowing for continuous exposure from the onset of the life cycle. To our knowledge, this is among the first studies to systematically evaluate dose–effect relationships in the context of the H. pluvialis life cycle, offering new perspectives on the optimal timing for nanoparticle application in astaxanthin-oriented biotechnological processes.
We have revised the Introduction, Materials and Methods, and Discussion sections to emphasize this contribution and better position our findings in relation to existing literature.
Reviewer Comment: H. pluvialis is an outdated and incorrect synonym; H. lacustris is the valid name (10.12705/652.11).
Response: We thank the reviewer for this taxonomic clarification. We are aware that Haematococcus lacustris is currently the valid name, according to Nakada & Ota (2016), as referenced in 10.12705/652.11. However, the strain used in this study is deposited and registered as Haematococcus pluvialis CNMN-AV-05, and this nomenclature continues to be widely used in applied and biotechnological literature. This situation is likely to persist, as in the case of Porphyridium cruentum and P. purpureum. A more established example is Spirulina (Arthrospira) platensis, which remains referred to as both a cyanobacterium and a microalga. This reflects the current status and dynamics of studies in systematics and nomenclature.
Reviewer Comment: The novelty is weak. There are several existing studies on the effects of TiOâ‚‚, ZnO, and CuO nanoparticles, including: 10.1016/j.jbiotec.2021.10.004, 10.7717/peerj.7582, 10.1142/S1088424624500664, and 10.1007/s10646-021-02406-5. Thus, the claim that this topic is poorly described (lines 63–66) lacks supporting evidence.
Response: We appreciate the reviewer’s observation and acknowledge that previous studies have investigated the effects of TiOâ‚‚, ZnO, and CuO nanoparticles on Haematococcus pluvialis. Among these, the study by Nasri et al. (2021) provides relevant data on the stimulation of astaxanthin biosynthesis by ZnO nanoparticles and has already been discussed in our manuscript. Additionally, studies such as those by Djearamane et al. (2019) and Babazadeh et al. (2021) report short-term cytotoxic effects (24–96 h) of ZnO and CuO nanoparticles under limited exposure conditions, without distinguishing the biochemical responses of the microalga. For Haematococcus, the physiological stage at which contact with nanoparticles occurs is particularly important, as cells undergoing active division are more sensitive to xenobiotics with toxic potential.
In contrast, our study was designed to assess the effects of these nanoparticles under continuous exposure, analyzing separately the three major developmental stages of H. pluvialis—green motile cells, palmella, and red aplanospores. This approach allows for stage-specific resolution combined with an analysis of the evolution of biomass, lipid content, and astaxanthin levels—features not previously reported in the cited literature.
We would also like to clarify that the current research context supports the statement in lines 63–66. While the general effects of nanoparticles on H. pluvialis have been explored, there is indeed a lack of studies specifically addressing their influence on the transition between developmental stages. Furthermore, many published studies do not clearly specify the type of inoculum used or the exact developmental stage at which nanoparticles were applied and their toxicity monitored, which limits reproducibility and biological interpretation.
The Introduction section has been updated to more clearly position the contribution of our study within the existing research landscape.
Reviewer Comment: The statistical treatment is inadequately described. Figure captions refer to ‘p<0.05; b – p<0.01; c – p<0.001’, but do not clarify the hypotheses tested or the specific comparisons made. In the Discussion (lines 310–313), correlation analysis is mentioned, yet no corresponding data or methodological description is provided.
Response: We thank the reviewer for this pertinent observation. The figure captions have been revised to specify that the indicated statistical significance refers to comparisons between each nanoparticle concentration and the control variant (0 mg/L), using Student’s t-test.
Additionally, the “Materials and Methods” section now includes an explicit statement of the tested hypotheses, indicating that a two-tailed t-test was applied for each treatment–control pair within each developmental stage.
The correlation analysis mentioned on line 310 was conducted using the Pearson coefficient, and the results were expressed as r² values.
It was our omission that the corresponding material was not included in the supplementary data.
Reviewer Comment: It is unclear from the Results section what exactly is represented by the bars in the plots. Do they depict differences between initial and maximum values? Or between initial and values obtained at a specific time point?
Response: We thank the reviewer for this observation. Each figure in the Results section corresponds to a measured parameter (biomass, astaxanthin, or lipids) for one or two specific developmental stages of the Haematococcus pluvialis life cycle. The figures specifically illustrate the dose-effect relationship of nanoparticles at defined life cycle stages.
The bars in the plots represent the absolute values of the measured parameters, obtained at specific sampling points for both treated and control cultures. These do not represent differences from initial values — all values reflect the direct results of analyses performed at each developmental stage.
The concept of “initial values” is not applicable in our experimental design, as the aim was not to compare temporal changes within a single culture, but to evaluate the direct effects of nanoparticles at well-defined stages of the life cycle. Each sampling point corresponds to a biologically distinct stage, and all comparisons were made between treated samples and the untreated control (0 mg/L NPs), collected simultaneously at the same developmental stage.
We would also like to note that the experiment was initiated with an inoculum consisting of red cysts (aplanospores), and nanoparticle treatments were applied at the beginning of cultivation. Therefore, our design is not based on comparisons at 24–72 hours post-exposure, but rather on the synchronized evaluation of biochemical responses at key developmental stages.
We have clarified this aspect in the figure captions and in the Results section to eliminate any ambiguity.
Reviewer Comment: The statements regarding astaxanthin ‘utilisation’ by algae (e.g. lines 21, 275, 280) should be reconsidered. Astaxanthin functions primarily as a passive photoprotective filter. There is currently no evidence for its degradation within algal cells. In some cases, a reduction in astaxanthin content as a percentage of biomass may reflect increased overall biomass, but this would require clearer explanation and detailed interpretation of the figures to support such a conclusion.
Response: We greatly appreciate the reviewer’s pertinent observation. We agree that the term “utilisation” is not appropriate in the context of astaxanthin metabolism in the absence of conclusive evidence. As correctly noted, astaxanthin primarily functions as a photoprotective and antioxidant molecule, accumulating in the cells.
We have revised the relevant statements in the manuscript to reflect this clarification. This adjustment also allows for a more accurate interpretation of differences between the various developmental stages of pigment content.
Reviewer Comment: The manuscript is poorly written, and the terminology used is often unclear. For instance, with respect to the haematocyst stage of the microalga, the terms ‘cyst’ and ‘aplanospore’ are used interchangeably, which adds to the confusion.
Response: We thank the reviewer for this comment. We understand the concern regarding the consistent use of terminology when describing the developmental stages of H. pluvialis, particularly with respect to the red stage. In the revised version of the manuscript, we have clarified the terminology and explicitly specified that the term “brown cyst” refers to the palmella stage, while “red cyst” corresponds to the aplanospore stage.
These stages are frequently referred to in the applied and research literature as “brown cyst” and “red cyst.”

Reviewer 2 Report
Comments and Suggestions for Authors
The paper describes the effects of TiO2, ZnO and CuO nanoparticles on astaxanthin accumulation in Haemacoccus pluvialis (HP). The influence of the nanoparticles was investigated at the palmella and aplanospore stage of HP. The employed particles are thought to intensify oxidative stress and, therefore, activate astaxanthin formation. The authors mention the complexity of the interaction between nanoparticles and microalgae such as HP. The paper investigates the influence of the particles on the life cycle of HP cells.
Increasing carotenoid formation from microalgae is not new, and the authors refer to previous publications. It would therefore have been more stimulating for a reader to stress the novelty of this manuscript early in abstract and introduction instead in line 64-65. Methods, experiments and results are clearly described. HP exhibited stage-specific responses to TiO2, ZnO and CuO nanoparticles.
The experiments result in significant facts: HP exhibited stage-specific responses to TiO2, ZnO and CuO nanoparticles. TiO2 particles enhance biomass content during the green phase with an increase of astaxanthin formation.
The paper is of good quality and can be published.
The authors may consider the following remarks as a suggestion.
It may be helpful for a non-specialized reader to give the molecular structure of astaxanthin and an illustration of the palmella and aplanospore stage of HP.
An interested reader, not specialized in this kind of research, might at first ask: When CuO has an effect, what effect exert the other Cu-oxid Cu2O? Then the question may emerge: why are the oxides of Ti, Zn, Cu used in this investigation? What about the oxides or salts of the neighbor elements Sc, Ni, Co, Fe, Mn, V? Even rare earth salts Ce3+, Gd3+, La3+ caused an increased formation of lutein, violaxanthin and b-carotene in microalgae, while Pr3+ and Lu3+ reduced the carotenoid formation. (F. Goecke et at., Effects of rare earth elements on growth rate, lipids, fatty acids and pigments in microalgae, Phycological Research 2017, 65, 226-234). Is there a more specific explanation beyond stress and ROS why some elements can increase the production of carotenoids?
Astaxanthin is a good antioxidant. Astaxanthin has been modified to increase the physiological properties by S. Lockwood and coworkers.
Astaxanthin for this research was predominantly synthesized by BASF. A good natural sources for astaxanthin would certainly be appreciated.
Retrometabolic Syntheses of Astaxanthin (3,3-dihydroxy-β,β-carotene-4,4-dione) Conjugates: A Novel Approach to Oral and Parenteral Cardioprotection.
S.F. Lockwood et al. Cardiovascular & Hematological Agents in Medicinal Chemistry, 2006, 4(4):335-49
The effects of oral Cardax (disodium disuccinate astaxanthin) on multiple independent oxidative stress markers in a mouse peritoneal inflammation model: influence on 5-lipoxygenase in vitro and in vivo
S.F. Lockwood, Life Sciences 79 (2006) 162 – 174
Author Response
We sincerely thank the reviewer for recognizing our work as the result of rigorous research and for expressing interest through thoughtful suggestions that may guide future studies.
Reviewer Comment: The paper describes the effects of TiO2, ZnO and CuO nanoparticles on astaxanthin accumulation in Haematococcus pluvialis (HP). The influence of the nanoparticles was investigated at the palmella and aplanospore stage of HP. The employed particles are thought to intensify oxidative stress and, therefore, activate astaxanthin formation. The authors mention the complexity of the interaction between nanoparticles and microalgae such as HP. The paper investigates the influence of the particles on the life cycle of HP cells.
Increasing carotenoid formation from microalgae is not new, and the authors refer to previous publications. It would therefore have been more stimulating for a reader to stress the novelty of this manuscript early in abstract and introduction instead in line 64-65. Methods, experiments and results are clearly described. HP exhibited stage-specific responses to TiO2, ZnO and CuO nanoparticles.
The experiments result in significant facts: HP exhibited stage-specific responses to TiO2, ZnO and CuO nanoparticles. TiO2 particles enhance biomass content during the green phase with an increase of astaxanthin formation.
The paper is of good quality and can be published.
Response: We thank the reviewer for the positive evaluation of our manuscript and the valuable suggestions. We have revised the Abstract and Introduction to highlight the novelty of our study more clearly from the beginning, as suggested. We appreciate the reviewer’s acknowledgment of the clarity of our methodology and the significance of the stage-specific responses observed in Haematococcus pluvialis under nanoparticle treatment.
Reviewer Comment: The authors may consider the following remarks as a suggestion. It may be helpful for a non-specialized reader to give the molecular structure of astaxanthin and an illustration of the palmella and aplanospore stage of HP.
Response: We sincerely thank the reviewer for this thoughtful and constructive suggestion. We fully agree that including the molecular structure of astaxanthin and visual representations of the palmella and aplanospore stages of Haematococcus pluvialis could be valuable, particularly for non-specialist readers.
However, we have opted not to include such figures in the present manuscript for the following reasons:
– While our study specifically targeted the physiological stages of H. pluvialis, we did not perform microscopic documentation or direct morphological characterization of the cells. As such, we believe it would not be appropriate to include illustrative images that were not generated as part of the current experimental work.
– With respect to the molecular structure of astaxanthin, although relevant in broader contexts, we did not explore its chemical behavior or structural properties in this study. Therefore, its inclusion was considered outside the scope of our investigation, which focused on biochemical responses to nanoparticle exposure.
To ensure clarity and preserve focus on the study’s primary objectives—namely, the stage-specific physiological and biochemical effects induced by nanoparticle treatments—we refrained from incorporating additional illustrative elements. This decision also aligns with our deliberate exclusion of a detailed analysis of the life cycle timing variations that may themselves be modulated by nanoparticle interaction.
Nevertheless, we sincerely appreciate the reviewer’s valuable insight and will consider integrating such visual elements in future publications with a more descriptive or educational emphasis.
Reviewer Comment: An interested reader, not specialized in this kind of research, might at first ask: When CuO has an effect, what effect exert the other Cu-oxid Cu2O? Then the question may emerge: why are the oxides of Ti, Zn, Cu used in this investigation? What about the oxides or salts of the neighbor elements Sc, Ni, Co, Fe, Mn, V? Even rare earth salts Ce3+, Gd3+, La3+ caused an increased formation of lutein, violaxanthin and b-carotene in microalgae, while Pr3+ and Lu3+ reduced the carotenoid formation. (F. Goecke et at., Effects of rare earth elements on growth rate, lipids, fatty acids and pigments in microalgae, Phycological Research 2017, 65, 226-234). Is there a more specific explanation beyond stress and ROS why some elements can increase the production of carotenoids?
Astaxanthin is a good antioxidant. Astaxanthin has been modified to increase the physiological properties by S. Lockwood and coworkers.
Astaxanthin for this research was predominantly synthesized by BASF. A good natural sources for astaxanthin would certainly be appreciated.
Retrometabolic Syntheses of Astaxanthin (3,3-dihydroxy-β,β-carotene-4,4-dione) Conjugates: A Novel Approach to Oral and Parenteral Cardioprotection. S.F. Lockwood et al. Cardiovascular & Hematological Agents in Medicinal Chemistry, 2006, 4(4):335-49
The effects of oral Cardax (disodium disuccinate astaxanthin) on multiple independent oxidative stress markers in a mouse peritoneal inflammation model: influence on 5-lipoxygenase in vitro and in vivo. S.F. Lockwood, Life Sciences 79 (2006) 162 – 174
Response: We sincerely thank the reviewer for this insightful and well-informed commentary. The observations regarding copper oxide speciation, the relevance of neighboring transition metals, and the potential of rare earth elements in modulating carotenoid synthesis are highly appreciated. We compared copper oxide nanoparticles with copper nanoparticles in their interaction with the cyanobacterium Spirulina platensis (10.3390/nano15010046).
Regarding rare earth elements, we have previously explored their biological effects and bioaccumulation in cyanobacteria, particularly in Spirulina platensis, and we would be pleased to cite several of our published studies in future work (10.3390/microorganisms12010122; 10.3390/cleantechnol5020032).
As for the stimulation of astaxanthin biosynthesis, other research groups have indeed investigated the use of coordination compounds of iron, zinc, and cobalt salts for their modulatory roles. We fully agree with the reviewer that mechanistic explanations exist beyond generalized oxidative stress or ROS involvement. In particular, the ability of specific metal ions to act as cofactors or structural elements in enzymes directly involved in the carotenoid biosynthesis pathway, including the conversion of β-carotene to astaxanthin, represents a more specific and biologically grounded mechanism (https://www.bioresearch.ro/2023-1/021-027-AUOFB.30.1.2023-RUDI.L.-Influence.of.Co%28II%29.compounds.pdf?utm_source=chatgpt.com).
While the current study focused on the stage-specific physiological responses of H. pluvialis to metal oxide nanoparticles, we acknowledge that a more in-depth exploration of metal-ion-specific enzymatic interactions would be an excellent direction for future research.
We thank the reviewer for highlighting the importance of natural astaxanthin sources, particularly in biomedical and nutraceutical applications. While our present study focuses on the endogenous production of astaxanthin by H. pluvialis under nanoparticle-induced stress, we agree that natural astaxanthin is highly interesting for future research.
In our previous work (including doctoral research), we have observed that natural astaxanthin exhibits greater long-term stability in vegetable oils and that these formulations showed increased thermal stress resistance (10.38045/ohrm.2024.4.01).
We are very grateful for the reviewer’s constructive suggestions and the valuable references shared, and we will consider these aspects in future applied research directions.

Reviewer 3 Report
Comments and Suggestions for Authors
The manuscript reported an interesting treatment of nanoparticles on the grwoth and Asta/lipid synthesis of microalga Haematococcus pluvialis. Results showed the effectiveness of the nanoparticles to enhance algal growth and Asta accumulation. Before it accepted , a minor revision is suggested as follows.
- As shown in the work, the effectiveness of nanoparticles is related with the life stage of Haematococcus pluvialis. In experimental methods, how to add the nanoparticels? For example, in green motile stage, the addition of nanoparticles resulted the increased growth (biomass ), then the algal cells entered into palmella and aplanospore stages, how about the biomass and Asta/Lipid change? As the same, when investigate the influence of nanoparticles added in palmella stage, how does the biomass and Asta content when the algal cells later enterred in aplanospore stage?
- Is there any influence of the particle size on the growth and Asta synthesis?
Author Response
Reviewer Comment: The manuscript reported an interesting treatment of nanoparticles on the grwoth and Asta/lipid synthesis of microalga Haematococcus pluvialis. Results showed the effectiveness of the nanoparticles to enhance algal growth and Asta accumulation. Before it accepted , a minor revision is suggested as follows.
As shown in the work, the effectiveness of nanoparticles is related with the life stage of Haematococcus pluvialis. In experimental methods, how to add the nanoparticels? For example, in green motile stage, the addition of nanoparticles resulted the increased growth (biomass ), then the algal cells entered into palmella and aplanospore stages, how about the biomass and Asta/Lipid change? As the same, when investigate the influence of nanoparticles added in palmella stage, how does the biomass and Asta content when the algal cells later enterred in aplanospore stage?
Response: We thank the reviewer for this insightful comment. The nanoparticles (TiOâ‚‚, ZnO, CuO) were added directly to the culture medium at the beginning of the experiment, simultaneously with the inoculation of H. pluvialis cultures, which consisted of red cysts (aplanospores). This experimental design allowed us to monitor the entire life cycle of the microalga under continuous exposure to nanoparticles.
Thus, the nanoparticle treatment began at the germination stage and persisted throughout the green motile, palmella, and aplanospore stages. Experimental samples were collected on specific days corresponding to each stage:
– Day 9, corresponding to the green motile stage
– Day 13, corresponding to the palmella stage
– Day 16, corresponding to the aplanospore stage
The samples collected at each of these stages underwent biochemical analyses, including biomass quantification, astaxanthin content, and lipid concentration.
As described in the Results section, biomass increased significantly during the green motile stage, particularly in cultures treated with TiOâ‚‚NPs. In the palmella stage, biomass accumulation continued, but astaxanthin levels showed a transient decrease, possibly due to its metabolic conversion or a temporary delay in biosynthesis. In the aplanospore stage, astaxanthin synthesis was reactivated, with a subsequent increase in pigment concentration, accompanied by a strong correlation with lipid content in this phase.
It is important to note that nanoparticles were not added separately at later stages (e.g., palmella), nor did we compare treatments initiated at different time points. Our experimental approach focused on the continuous presence of nanoparticles from the onset of cultivation, allowing us to assess their cumulative and stage-specific metabolic effects.
We have revised the “Materials and Methods” section accordingly and provided additional clarification in the Results and Discussion to better communicate our methodology and its implications.
Reviewer Comment: Is there any influence of the particle size on the growth and Asta synthesis?
Response: We thank the reviewer for this relevant question. In the present study, we used commercially available metal oxide nanoparticles (TiOâ‚‚, ZnO, CuO) with predefined average particle sizes as specified by the manufacturers. While we acknowledge that particle size can significantly influence cellular uptake, surface reactivity, and ROS generation — all of which may impact microalgal growth and astaxanthin synthesis — our experimental design did not include a comparison of different particle sizes.
This study aimed to evaluate the stage-specific physiological responses of Haematococcus pluvialis to a fixed nanoparticle type and concentration. Therefore, any influence of particle size was beyond the scope of the current investigation.
We agree that investigating the effect of nanoparticle size on carotenoid biosynthesis would be an important and valuable extension of this research, and we appreciate the reviewer’s suggestion in this regard.

Round 2
Reviewer 1 Report
Comments and Suggestions for Authors
Unfortunately, the authors have approached the revision formally. The most serious issues remain unresolved.
H. pluvialis is an outdated and incorrect synonym; H. lacustris is the valid name (see 10.12705/652.11; AlgaeBase: https://www.algaebase.org/search/species/detail/?species_id=27370; 10.4490/algae.2023.38.3.9; NCBI Taxonomy: https://www.ncbi.nlm.nih.gov/Taxonomy/Browser/wwwtax.cgi?id=44745). Unfortunately, this issue has not been resolved.
The novelty is weak and has not been justified. Both vegetative growth and astaxanthin accumulation during the transition to the aplanospore (haematocyst) stage have already been documented (see references from the first review round). This is merely a case study suited to a more specialised journal.
The statistical treatment is inadequately described. Based on the data presentation, I cannot assess the reliability of the conclusions or the results in each section. The description of the experiments in the Results section is confusing and incomprehensible. It is unclear whether the authors gradually increased nanoparticle concentration during algal cultivation (lines 136–137).
"Content remained within normal variation limits" (line 155) — what are these normal variation limits, and how were they determined?
Lines 292–293: "Astaxanthin, predominantly synthesised in response to stress, was significantly reduced during the palmella stage" — reduced during which process? Under what conditions? What is the control?
The control has not been clearly defined. The authors state it is 0 mg/ml of alga (lines 81, 102, 133, 164), i.e., “0 – control variant” (lines 80, 101, 132, 163) — but 0 of what?
Lines 430–431: Which groups are being compared? Student’s t-test is used for pairs of normally distributed data, yet I observe more than two bars in all diagrams. The authors also mention correlation analysis, but no correlation data are presented in the results. Details of this analysis have likewise not been described (lines 431–434). I see no p values or Pearson r values in the text.
Furthermore, there is a methodological issue: determining optical density at the listed wavelengths more accurately reflects intracellular pigment content than culture dry mass (line 380) (see 10.1007/s00253-012-4677-9).
The terminology used is often unclear. For instance, in reference to the haematocyst stage of the microalga, the terms “cyst” and “aplanospore” are used interchangeably, which adds to the confusion (e.g., lines 19, 67, 84, 97, 101, 108, 118, 126, 212, 243, 278, 281, 333, 356, 363, 364, 386, 387, 391). I am afraid the authors are not familiar with the current terminology regarding the Haematococcus life cycle (see 10.3389/fpls.2016.00531; 10.3390/md21020108). It also appears they do not distinguish between the terms “stage” and “phase” (e.g., line 315).
Author Response
Reviewer Comment: Unfortunately, the authors have approached the revision formally. The most serious issues remain unresolved.H. pluvialis is an outdated and incorrect synonym; H. lacustris is the valid name (see 10.12705/652.11; AlgaeBase: https://www.algaebase.org/search/species/detail/?species_id=27370; 10.4490/algae.2023.38.3.9; NCBI Taxonomy: ttps://www.ncbi.nlm.nih.gov/Taxonomy/Browser/wwwtax.cgi?id=44745). Unfortunately, this issue has not been resolved. Thank you for the observation.
Response: The name Haematococcus pluvialis has been replaced with Haematococcus lacustris throughout the manuscript, in accordance with current taxonomic standards. We agree that promoting the correct nomenclature is important, especially in scientific publications.
Reviewer Comment: The novelty is weak and has not been justified. Both vegetative growth and astaxanthin accumulation during the transition to the aplanospore (haematocyst) stage have already been documented (see references from the first review round). This is merely a case study suited to a more specialised journal.
Response: Thank you for your comment. We agree that the general aspects of vegetative growth and astaxanthin accumulation during the transition to the aplanospore stage have been previously described in the literature. However, we believe that the novelty of our study lies in the stage-specific evaluation of the effects of TiOâ‚‚, ZnO, and CuO nanoparticles, which were applied from the start of cultivation and monitored throughout the entire life cycle of Haematococcus lacustris.
To our knowledge, a comparative analysis of physiological responses (biomass, astaxanthin, and lipids) across the three distinct stages—motile, palmella, and aplanospore—in the presence of these nanoparticles has not been previously detailed in this manner. We therefore consider that our study makes a valuable contribution to understanding the timing and impact of nanomaterial application in algal biotechnology. While it may not introduce a completely new research direction, it offers an in-depth and targeted analysis of a documented biological process.
We have reviewed the studies suggested as similar, and although they share certain elements with our work, they differ significantly in terms of objectives, experimental design, and analysis. Therefore, we consider them complementary, but not equivalent to our study.
10.1016/j.jbiotec.2021.10.004. It is a comparable but not equivalent study. It supports the idea that ZnO nanoparticles can stimulate astaxanthin production within a specific concentration range; however, it does not address the dynamics of the life cycle, does not investigate other metabolites, and does not analyze stage-specific metabolic changes. Therefore, although it may seem to partially diminish the novelty of our study, it actually complements it by reinforcing the relevance of nanoparticle applications in microalgal biotechnology.
10.7717/peerj.7582. This study investigated the toxic effects of ZnO nanoparticles on Haematococcus pluvialis, focusing on cellular accumulation, growth inhibition, cell death, and the reduction of photosynthetic pigments over short exposure periods (24–96 h). While it provides important insights for ecotoxicological risk assessment, this study does not examine stage-specific physiological responses of the microalga, nor does it explore the potential application of nanoparticles in a biotechnological context. In contrast, our work investigates the stage-specific physiological effects of ZnO, TiOâ‚‚, and CuO nanoparticles on Haematococcus lacustris, with the aim of identifying optimal conditions for enhancing biomass and astaxanthin productivity.
10.1142/S1088424624500664. This study applied Haematococcus pluvialis cells as a biological matrix for preparing TiOâ‚‚-based photoelectrocatalytic composites, aiming to enhance COâ‚‚ reduction efficiency through dye-sensitization by astaxanthin. While it involves both H. pluvialis and TiOâ‚‚ nanoparticles, the focus is on materials engineering and photocatalytic performance, rather than on the biological effects of nanoparticles on algal physiology or astaxanthin biosynthesis.
10.1007/s10646-021-02406-5. This study evaluated the toxicological impact of CuO/perlite nanoparticles on Haematococcus pluvialis, focusing on short-term effects (up to 96 hours). Although astaxanthin content is reported, the analysis is limited to a single exposure period, without monitoring a complete algal life cycle and without differentiation between physiological stages. We therefore consider this study to be complementary, but not equivalent.
Reviewer Comment: The statistical treatment is inadequately described. Based on the data presentation, I cannot assess the reliability of the conclusions or the results in each section. The description of the experiments in the Results section is confusing and incomprehensible. It is unclear whether the authors gradually increased nanoparticle concentration during algal cultivation (lines 136–137).
Response: We thank the reviewer for this important observation. We confirm that nanoparticles were applied once at the start of cultivation, and no gradual increase in concentration was performed during the experimental period. Each tested concentration was compared exclusively with the control group (0 mg/L) using Student’s t-test, and statistical significance is clearly indicated in the figure legends. Moreover, the figures distinguish the nanoparticles, and the x-axis specifies their concentrations, ranging from 0 mg/L (control culture, without nanoparticles) to 30 mg/L, depending on the experimental variant. To improve clarity, we have added a detailed description of the experimental design in the Materials and Methods section.
Reviewer Comment: "Content remained within normal variation limits" (line 155) — what are these normal variation limits, and how were they determined?
Response: We thank the reviewer for the observation. The sentence has been revised in the manuscript to more accurately reflect the observed effects on astaxanthin content and to clarify that the values were either comparable to or higher than those of the control.
Reviewer Comment: Lines 292–293: "Astaxanthin, predominantly synthesised in response to stress, was significantly reduced during the palmella stage" — reduced during which process? Under what conditions? What is the control?
Response: We thank the reviewer for the observation. The sentence in question has been removed, and the discussion section has been revised to clarify the impact of the nanoparticles. In any case, the original sentence referred to a comparison with the control variant, without nanoparticles.
Reviewer Comment: The control has not been clearly defined. The authors state it is 0 mg/ml of alga (lines 81, 102, 133, 164), i.e., “0 – control variant” (lines 80, 101, 132, 163) — but 0 of what?
Response: We thank the reviewer for the observation. The mention “0 – control variant” refers to the nanoparticle concentration (0 mg/L), not to the absence of algal biomass. This is indicated on the X-axis of the figures, where nanoparticle concentrations are shown ranging from 0 mg/L to 30 mg/L. To avoid any confusion, we have clarified this aspect in the description of the experimental design in the “Materials and Methods” section.
Reviewer Comment: Lines 430–431: Which groups are being compared? Student’s t-test is used for pairs of normally distributed data, yet I observe more than two bars in all diagrams. The authors also mention correlation analysis, but no correlation data are presented in the results. Details of this analysis have likewise not been described (lines 431–434). I see no p values or Pearson r values in the text.
Response: We thank the reviewer for the comments. We confirm that in all cases, Student’s t-test was used for pairwise comparisons between each treatment (nanoparticle concentration) and the control variant (0 mg/L), in accordance with the method’s requirements. This was applied individually for each experimental group (n = 3) and is reflected in the figure legends through the indication of statistical significance relative to the control.
Regarding the correlation analysis, it was performed and the results were included in the supplementary material (Table S1); however, the corresponding reference was not initially included in the main text. This omission has been corrected by adding an explicit reference to the supplementary table in the Discussion section.
Reviewer Comment: Furthermore, there is a methodological issue: determining optical density at the listed wavelengths more accurately reflects intracellular pigment content than culture dry mass (line 380) (see 10.1007/s00253-012-4677-9).
Response: We thank the reviewer for the comment. This is not a methodological error. Spectrophotometric measurement of optical density at specific wavelengths, correlated with gravimetric data, is a well-established method for the rapid and reliable quantification of microalgal biomass (10.3390/en16052429; 10.1016/j.aquaculture.2015.05.044). These methods, based on OD–dry weight calibration curves, are widely used in experiments involving a large number of samples and provide an efficient alternative to direct gravimetric determinations. Haematococcus is no exception, as supported by the literature (e.g., 10.1186/s13068-025-02604-x).
The study cited by the reviewer (10.1007/s00253-012-4677-9) focuses on the nondestructive estimation of pigment ratios and carotenoid content under stress conditions, using normalized spectral indices derived from multiple wavelengths. In that context, relying on a single wavelength is not sufficient for accurately capturing pigment composition.
Reviewer Comment: The terminology used is often unclear. For instance, in reference to the haematocyst stage of the microalga, the terms “cyst” and “aplanospore” are used interchangeably, which adds to the confusion (e.g., lines 19, 67, 84, 97, 101, 108, 118, 126, 212, 243, 278, 281, 333, 356, 363, 364, 386, 387, 391). I am afraid the authors are not familiar with the current terminology regarding the Haematococcus life cycle (see 10.3389/fpls.2016.00531; 10.3390/md21020108). It also appears they do not distinguish between the terms “stage” and “phase” (e.g., line 315).
Response: We thank the reviewer for the observation. We have revised the manuscript and replaced the term “cyst” to avoid potential ambiguity. We have also replaced the term “phase” with “stage” to maintain terminological consistency.
